# Sub-10-fs control of dissociation pathways in the hydrogen molecular ion with a few-pulse attosecond pulse train

Yasuo Nabekawa[1], Yusuke Furukawa[1,†], Tomoya Okino[1], A. Amani Eilanlou[1], Eiji J. Takahashi[1], Kaoru Yamanouchi[1,2] & Katsumi Midorikawa[1]

The control of the electronic states of a hydrogen molecular ion by photoexcitation is considerably difficult because it requires multiple sub-10 fs light pulses in the extreme ultraviolet (XUV) wavelength region with a sufficiently high intensity. Here, we demonstrate the control of the dissociation pathway originating from the $2p\sigma_u$ electronic state against that originating from the $2p\pi_u$ electronic state in a hydrogen molecular ion by using a pair of attosecond pulse trains in the XUV wavelength region with a train-envelope duration of $\sim 4$ fs. The switching time from the peak to the valley in the oscillation caused by the vibrational wavepacket motion in the $1s\sigma_g$ ground electronic state is only 8 fs. This result can be classified as the fastest control, to the best of our knowledge, of a molecular reaction in the simplest molecule on the basis of the XUV-pump and XUV-probe scheme.

[1] Attosecond Science Research Team, Extreme Photonics Research Group, RIKEN Center for Advanced Photonics, 2-1 Hirosawa, Wako-shi, Saitama 351-0198, Japan. [2] Department of Chemistry, School of Science, the University of Tokyo, 7-3-1 Hongo, Bunkyo-ku, Tokyo 113-0033, Japan. † Present address: Department of Engineering Science, the University of Electro-Communications, 1-5-1 Chofugaoka, Chofu, Tokyo 182-8585, Japan. Correspondence and requests for materials should be addressed to Y.N. (email: nabekawa@riken.jp).

The progress of laser light sources towards a shorter pulse duration in the past 20 years[1–13] has been beneficial for investigating the ultrafast dynamics of atoms and molecules, in which the time scale of molecular state evolution typically ranges from 100 as to 10 fs. In the measurement of such ultrafast dynamics, we generally use a pair of ultrashort pulses one of which is utilized for initiating the state evolution of a target molecule as a pump. The other pulse, used as a probe, is irradiated after a delay relative to the pump pulse and causes the target molecule to change to another state so that an observable, such as the transient absorbance in the extreme ultraviolet (XUV) region[14–16], the yields or kinetic energy of ions[17–24], or the kinetic energy or angular distribution of continuum electrons[17,25–31], can reveal a trace of the state evolving at the time of delay. The temporal resolution of this kind of pump-probe measurement can be finer than 100 as owing to the mature technologies for generating an isolated attosecond pulse (IAP) originating from an XUV high harmonic of a-few-cycle near-infrared pulse with carrier envelope phase stabilization. We can utilize the near-infrared pulse as a pump pulse and the IAP as a probe or vice versa owing to the complete phase locking between the two pulses as the intrinsic nature of the high-harmonic (HH) generation process.

From the point of view of the target molecular system to be observed, the hydrogen molecule ($H_2$) and ion ($H_2^+$) have been attracting great interest from researchers in molecular/attosecond physics. This is because their simplest structures consisting of two protons and one ($H_2^+$) or two ($H_2$) electrons should provide benchmarks for the fundamental characteristics of electronic[20,21] and nuclear dynamics[19,32,33] in the sub-femtosecond and 10-fs regimes, respectively. In fact, XUV IAP pump and near-infrared probe experiments have revealed the vibrational motion of $H_2^+$ and the time evolution of the coherent superposition of multiple electronic states in $H_2^+$ (refs 19,20,34). An XUV attosecond pulse train (APT) composed of discrete HH components has also been useful for investigating the electronic states in $H_2$ and $H_2^+$ (refs 21,24).

In spite of these successful studies on ultrafast dynamics in $H_2$ and $H_2^+$ based on XUV pump and near-infrared probe measurement, we should focus more attention on the advantages of XUV pump and XUV probe measurements as pointed out by Palacios et al. in ref. 35. They claim that an XUV probe pulse can be used to explore field-free molecular dynamics without distorting the adiabatic potential of a molecule owing to the negligible ponderomotive energy given to electrons upon exposure to an XUV field. Our previous works on probing the nuclear dynamics of $D_2^+$ and $H_2^+$ using ultraviolet and vacuum ultraviolet pulses[36–38] are grounded on this fact because we assume that the adiabatic potentials are not deformed by the probe pulses in our analysis of experimental data. In addition to this benefit, the use of an XUV IAP/APT extends the control scheme of chemical reactions in femtochemistry[39,40], which typically utilizes the sub-picosecond nuclear dynamics of excited electronic states in a heavy molecule with an excitation energy of several eVs to the control scheme of chemical reactions in the 10 fs or even attosecond regime. This is because the high photon energy of the XUV IAP/APT of more than 10 eV and its broad photon energy bandwidth of more than 5 eV make it possible to simultaneously excite multiple electronic states to form an attosecond electronic wavepacket in a light molecule with ∼10 fs nuclear wavepacket evolution as was demonstrated in ref. 41.

In this paper, we report on the control of the two dissociation pathways of the lightest molecular ion, $H_2^+$, by irradiating a two-pulse APT sequence, the duration of the train envelope of which is estimated to be ∼4 fs. The first APT in the sequence ionizes a $H_2$ molecule by absorbing one photon in the APT and launches the vibrational wavepacket into the $1s\sigma_g$ electronic state of the $H_2^+$ molecular ion. The second APT leads to the one-photon transition from the $1s\sigma_g$ state to the $2p\sigma_u$ and $2p\pi_u$ states, resulting in dissociation. The transition amplitude from the $1s\sigma_g$ state is altered by the motion of the vibrational wavepacket and differs for each repulsive potential curve of the relevant electronic state. Therefore, we can maximize and minimize the $H^+$ fragment yield via the $2p\sigma_u$ state and that via the $2p\pi_u$ state, respectively, and vice versa, by adjusting the delay of the second pulse to be near the time of the half-revival of the vibrational wavepacket. Switching from the $2p\sigma_u$-rich $H^+$ yield to the $2p\pi_u$-rich $H^+$ yield occurs during the half-period of the vibrational motion, resulting in ∼8 fs control of the dissociation pathways. The final product of the $2p\sigma_u$ state is $H^+ + H(1s)$, while that of the $2p\pi_u$ state is $H^+ + H(2s)$. Hence, our experimental result can be regarded as the simplest reaction control in accordance with the method proposed by Tannor, Kosloff and Rice[42,43].

## Results

**Velocity map image and assignment of dissociation pathways.** The experiment was performed based on the measurement of the $H^+$ fragment ions using a velocity map imaging (VMI) spectrometer. The image of $H^+$ fragment ions were recorded with scanning delay between the APT pump and APT probe. The experimental details are described in Methods section. We show an image of $H^+$ fragment ions obtained by counting the light spots in the acquired images at all delay positions in Fig. 1a. The colour scale of the left-hand side of this image expresses the entire signal range of the image, while that of the right-hand side is enhanced 10-fold. The direction of the polarization of the APT is indicated by the arrow labelled $E_{APT}$. We symmetrized the raw image to improve the signal-to-noise (S/N) ratio.

We retrieved the velocity map image of the $H^+$ fragment ion sliced on a plane parallel to the detection plane of the micro-channel plate (MCP) by applying the polar basis set expansion (pBasex)[44] Abel transform method. The resultant image is shown in the left-hand side of Fig. 1b. We compensate for the low signals of the outer rings by multiplying square of velocity, resulting in the image on the right-hand side of this figure. This correction of the signal is equivalent to the integration of the ion signal with respect to the solid angle as described in ref. 44. We define the $x$ direction to be perpendicular and the $z$ direction to be parallel to the polarization direction of the APT.

We have found that six pairs (half) of the crescent-shaped spectra representing the ion yield are arranged parallel to the polarization direction of the APT ($z$ direction) in the right-hand side of Fig. 1b. The spectra around the lowest three peak kinetic energies (KEs) at 0.44, 1.49 and 2.79 eV are assigned to be composed of the $H^+$ fragments originating from the $2p\sigma_u$ state excited by one-photon absorption of the fundamental, 3rd-harmonic and 5th-harmonic components in the APT, respectively, as we have already demonstrated in previous studies[36–38]. The new findings in the current study are the outer crescent-shaped spectra around the KEs of 4.45, 5.92 and 7.74 eV.

Note that we calibrated the kinetic energy release (KER), which is twice the KE, in accordance with the manner described in Supplementary Note 3 in ref. 38. The relation between the pixel number and the KER is depicted in Supplementary Fig. 1.

Another noticeable feature in the right-hand side of Fig. 1b is the existence of $H^+$ spectra biased perpendicularly to the polarization direction of the APT ($x$ direction), shown in three

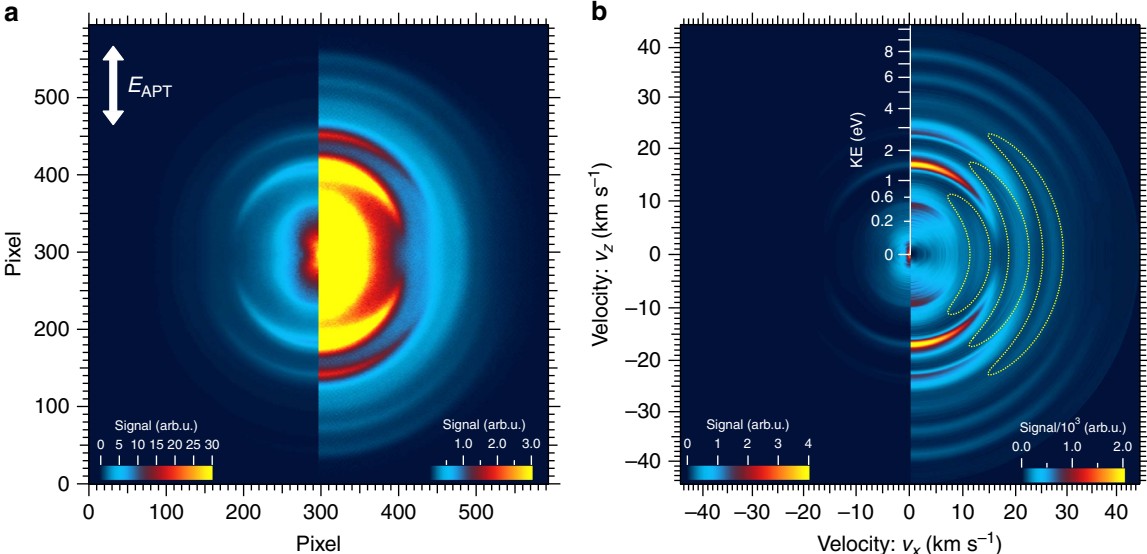

**Figure 1 | Images of H$^+$ fragment.** (**a**) Image of H$^+$ fragment obtained from the accumulation of all the acquired data. We extracted each image with a size of 595 pixels × 595 pixels from each image data recorded by the sCMOS camera such that the centre of the fragment image is located at pixel [297,297], where the pixel number is started from 0 in the extracted image. The left-hand side of the figure is the image depicted with a colour scale spanning the whole range of the signal. The colour scale is 10-fold magnified in the right-hand side of this figure to clearly exhibit the crescent images of the fragment with low signals. The direction of polarization of the attosecond pulse train (APT) is indicated by an arrow. (**b**) Left-hand side: velocity map image of H$^+$ fragment sliced on a plane parallel to the MCP, which is retrieved by using the pBasex method for image inversion[44]. Right-hand side: velocity map image with simulated polar binning obtained by multiplying square of velocity by the signal of the image on the left-hand side of this figure in accordance with the method described in ref. 44. The axis presenting the KE of the H$^+$ fragment is superimposed.

dotted contours, which have not been previously seen in our experiments. We show the KER spectra of H$^+$ fragments along the $z$ direction and $x$ direction in Fig. 2 as a fine blue curve with a shaded area and a thick red curve, respectively, to clearly exhibit the new fragment components as distinct peaks, which are labelled with their harmonic orders. The relationship between the peaks and the harmonic orders will be explained later. The line profiles are obtained from the right-hand-side image of Fig. 1b and the KER is double the KE.

We determined how the newly found H$^+$ spectra are generated as follows. It is natural to assume that the H$^+$ spectra biased in the $z$ direction originate from the $2p\sigma_u$ state because of the origin of the known spectra. We can also expect that the high KEs of the H$^+$ spectra are due to the high photon energies absorbed in the excitation process. Thus, we suppose that the excitation scheme shown in Fig. 3a is reasonable. In this scheme, the $1s\sigma_g$ state in the H$_2^+$ molecule is efficiently excited into the $2p\sigma_u$ state at the nuclear distance where the energy gap between the two states coincides with the photon energy of a harmonic component in the APT. The peak KER of the H and H$^+$ system is specified by the energy of the $2p\sigma_u$ state measured from the dissociation limit at this nuclear distance.

The KER peaks of the thin blue curve with the shaded area in Fig. 2 are in good agreement with those expected from this supposition. In addition, the KER spectrum calculated from a theoretical model[37,38] based on this supposition, which is depicted as a thin black curve with a shaded area on the right axis of Fig. 3, reproduces the measured spectrum with the reasonable KER peak positions (the measured spectrum are compared with the calculated spectrum in Supplementary Fig. 2a). Thus, we are convinced that the 'parallel' KER spectrum can be ascribed to the dissociation pathway schematically shown in Fig. 3a, and we attach each tag showing the harmonic order contributing to the excitation corresponding to each peak of the thin blue curve with the shaded area in Fig. 2.

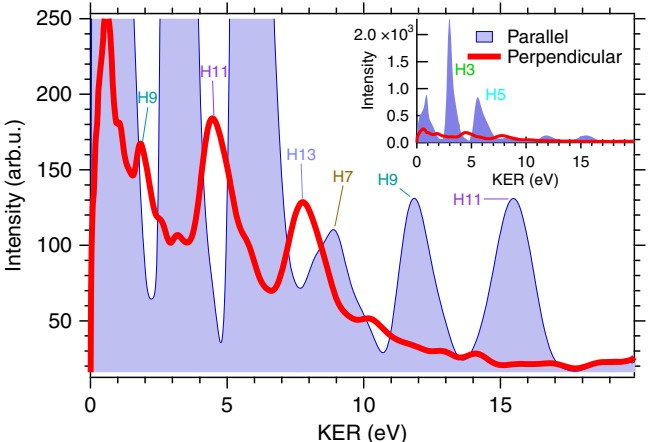

**Figure 2 | Kinetic energy release (KER) spectra of H$^+$ fragments.** KER spectra of H$^+$ fragments parallel ($z$ direction) and perpendicular ($x$ direction) to the polarization direction of the APT are depicted as fine blue curve with shaded area and red curve, respectively. The traces of the spectra are obtained from the image on the right-hand side of Fig. 1b. A magnified view is shown in the main panel, while the entire view is shown in the inset.

The origin of the H$^+$ fragments biased in the direction perpendicular to the APT polarization direction is expected to be the dissociation in an electronic state with $\pi_u$ symmetry. Thus, we have assumed that the $2p\pi_u$ state is the excited state generating the 'perpendicular' KER spectrum of H$^+$ fragments because this state exhibits the lowest adiabatic potential energy curve in the $\pi_u$ symmetry[45]. The three peak energies observed in the KER spectrum are the same as those expected from the correspondence of the photon energies of the 9th-, 11th- and 13th-harmonic

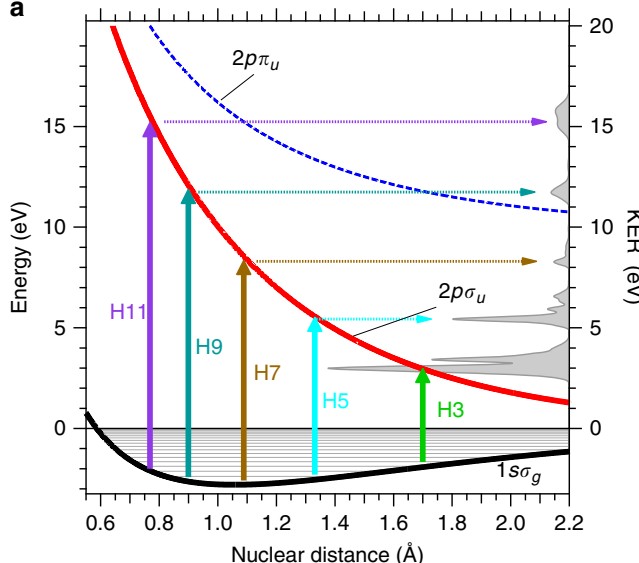

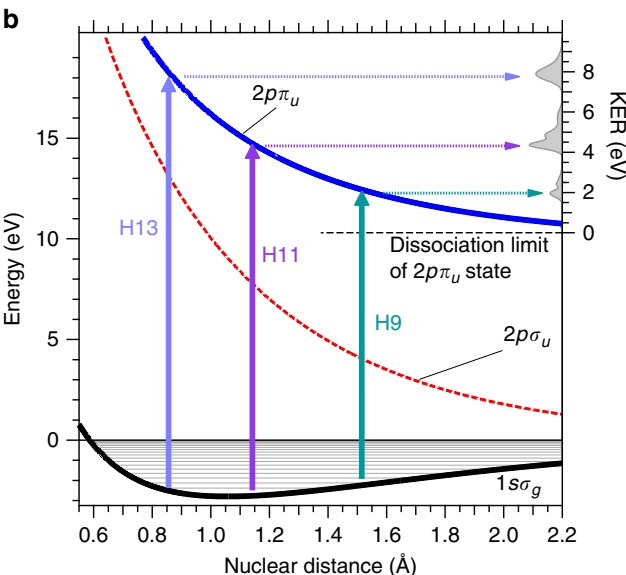

**Figure 3 | Excitation schemes in the $H_2^+$ molecular ion. (a)** Schematic figure of the one-photon excitation from the ground $1s\sigma_g$ state to the repulsive $2p\sigma_u$ state in the $H_2^+$ molecular ion. Black and red solid curves are the adiabatic potentials of the $1s\sigma_g$ and $2p\sigma_u$ states, respectively. The line profile with the shaded area on the right axis is the KER spectrum of a $H^+$ fragment calculated from the model described in the text. Horizontal lines depicted in the $1s\sigma_g$ potential well indicate the vibrational energy levels of nuclei. The length of each arrow represents the photon energy of each harmonic component in the APT. The order of each harmonic is shown by each arrow. **(b)** Schematic figure of the one-photon excitation from the ground $1s\sigma_g$ state to the repulsive $2p\pi_u$ state in the $H_2^+$ molecular ion. The adiabatic potentials, the KER spectrum of the $H^+$ fragment and other features are shown with in a similar manner to those depicted in (**a**).

components with the energy gap between the $1s\sigma_g$ and $2p\pi_u$ states, and the calculated KER spectrum shown on the right axis in Fig. 3b is in good agreement with the observed KER spectrum (the measured spectrum are compared with the calculated spectrum in Supplementary Fig. 2b). Therefore, we determine that the 'perpendicular' KER spectrum originates from the dissociation via the $2p\pi_u$ state and we label the three distinct peaks in the red curve in Fig. 2 with the harmonic order contributing to the yield of the

$H^+$ fragments for each peak. We neglect the excited states other than the $2p\sigma_u$ and $2p\pi_u$ states in our analysis for simplicity, although the measured KER spectrograms may contain minor contributions from highly excited states that are not taken into account.

**Delay–KER spectrograms**. We have analysed each recorded $H^+$-counting image at each delay with the pBasex method, and the KER spectra parallel and perpendicular to the polarization direction of the APT are arranged in accordance with the delay. The resultant delay-KER spectrograms (DKS) for the parallel direction and that for the perpendicular direction are shown in Fig. 4a,b, respectively. Note that we have corrected the gradual decrease in the intensity of the signal with increasing the delay, which is attributed to the degradation of the silicon harmonic separator mirror (SiBS)[46] during long-time data acquisition and the pitch and yaw of the piezo-translation stage, by the method described in Supplementary Note 3 in ref. 38. We label each harmonic order contributing to each KER component on the right-hand side of each figure.

We can clearly see oscillations of the H3 KER component around 3 eV and the H5 KER component around 5.6 eV as a function of the scanning delay in Fig. 4a. This is owing to the fact that the time evolution of the vibrational wavepacket created in the $1s\sigma_g$ state by the irradiation of the preceding APT with one-photon ionization is mapped onto the $2p\sigma_u$ state with the excitation accompanied by one-photon absorption of the 3rd- and 5th-harmonic components in the delayed APT. We also find that the H9 and H11 KER components oscillate at delay times of around 0 and 280 fs in a similar manner to the oscillation found in the H3 and H5 components, while there seems to be no clear oscillation at other delay times for the H9 and H11 KER components and in the whole range of the delay times for the H7 KER component.

We have calculated the DKS by following the theoretical model described in refs 37,38 to determine whether or not the characteristics in the measured DKS mentioned above are reasonable. The magnitude of each harmonic component in the APT is adjusted so as to reproduce the height ratio among the KER peaks in the measured DKS, and the chirp of the APT is fixed to $0.7 \times 10^{-32}$ s$^2$ in accordance with the result in ref. 38. The resultant DKS is shown in Fig. 4c.

The H9 and H11 KER components exhibit distinct peaks across their KER ranges at delay times of around 0, 280 and 560 fs, and exhibit complex textures with low intensities at other delay times. This is because the vibrational wavepacket, which is spatially localized at the initial time, is increasingly dispersed owing to the anharmonicity of the vibrational energies, and then it is localized again at approximately the time of the half-revival of 280 fs and its integer multiples. These features in the H9 and H11 components are consistent with those revealed in the experimental DKS in Fig. 4a except for the unclear oscillation around at 560 fs in the experimental DKS, which is ascribed to the degraded spatial overlap between the (ionizing) XUV components of the pump in the precedent APT and the 9th- and 11th-harmonic components of the probe in the delayed APT caused by the pitch and yaw of the translation stage. The degradation of the spatial overlap is reduced when the probe pulse is the 3rd- or 5th-harmonic component with a large beam diameter at the focal point, resulting in clear oscillations around 580 fs in the H3 and H5 KER components.

The signal intensity of the H7 KER component in the calculated DKS in Fig. 4c is considerably low due to the low intensity of the 7th-harmonic component in the APT. The magnitude of the oscillation is also low even when the delay

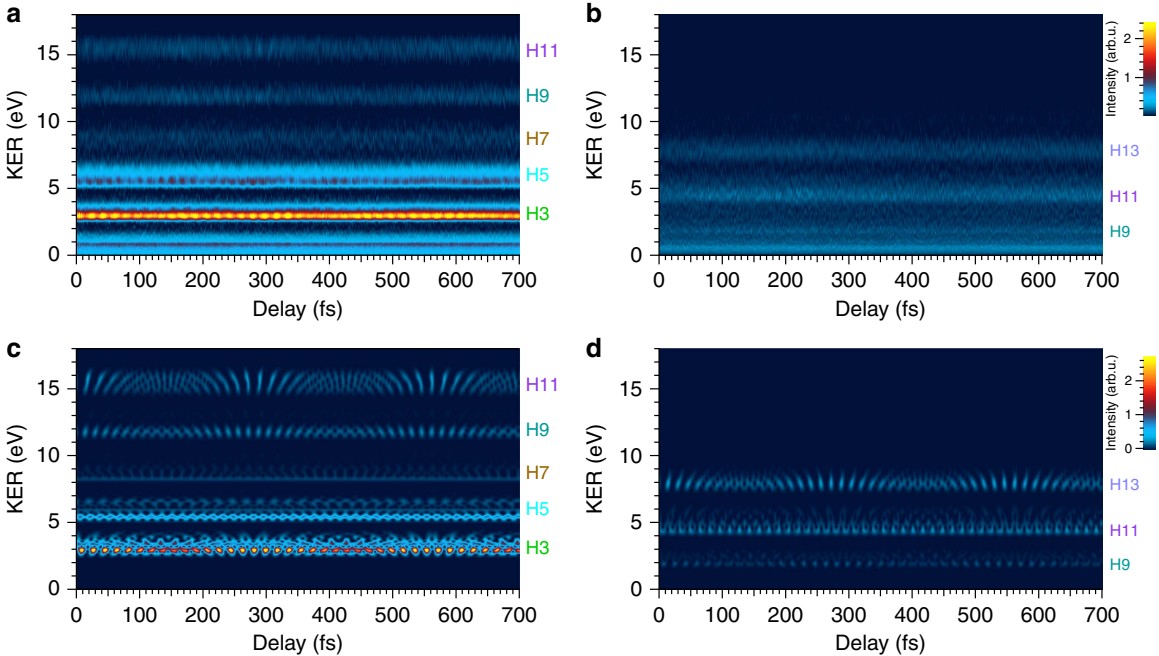

**Figure 4 | Delay–KER spectrograms (DKSs) of the H$^+$ fragment.** (**a**) Experimental result of DKS obtained from the parallel component of the sliced images. (**b**) Experimental result of DKS obtained from the perpendicular component of the sliced images. The colour scales indicating the intensities in (**a**) and (**b**) are common. (**c**) DKS of the H$^+$ fragment produced via the $2p\sigma_u$ state calculated from a theoretical model. (**d**) DKS of the H$^+$ fragment produced via the $2p\pi_u$ state calculated from a theoretical model. The colour scales indicating the intensities in (**c**) and (**d**) are common. We show the number of each harmonic order contributing to each KER component on the right-hand side of each figure.

reaches integer multiples of the time of the half-revival because the excitation from the $1s\sigma_g$ state to the $2p\sigma_u$ state mainly occurs when the vibrational wavepacket propagates across the equilibrium distance, as shown in Fig. 3a, with the maximum velocity. These are also good reasons for the vague oscillation of the H7 component in the experimental DKS in Fig. 4a.

More specific evidence for the vibrational wavepacket is revealed by resolving the frequency components of the oscillation. We show the magnitude square of the Fourier transforms of the DKSs in Fig. 4a–d in Fig. 5a–d, respectively, which we call frequency–KER spectrograms (FKSs). In Fig. 5a, obtained from the experiment, we find that the distinct peaks appearing at the difference frequencies between adjacent vibrational states, which are marked with the parentheses $(v, v + 1)$ on the top axis, are in reasonable agreement with those appearing in Fig. 5c obtained from the theoretical model. We indicate the regions where the difference frequencies appear in Fig. 5c by the red dashed contours in Fig. 5a.

Most of the peaks at the difference frequencies between the next-adjacent vibrational states, which are assigned by the parentheses $(v, v + 2)$ on the top axis, and some of the adjacent difference-frequency peaks are not so clear due to the insufficient S/N ratio of the recorded data. Nevertheless, the clearly visible difference-frequency peaks in the H9 ((0,1), (1,2) and (2,3)) and H11 ((2,3), (3,4), (4,5), (5,6) and (6,7)) KER components are clear evidence that the time evolution of the vibrational wavepacket modulates the H$^+$ yields in these KER components.

We can recognize from Fig. 5c that the intensity variation in the DKS is determined by only the discrete difference-frequency components in the FKS, and thus the frequency components in other regions found in Fig. 5a should be specified as noises. We rejected these frequency noises by applying a band-pass filter (BPF) to the complex amplitude of the Fourier transform of the DKS in Fig. 4a, and then we carried out the inverse Fourier transform. The passbands of the filter in the frequency–KER

domain are depicted with red dashed contours showing 40% of the maximum intensity of the super-Gaussian filter in Fig. 5a. They pass all the difference-frequency components of $(v, v + 1)$ and $(v, v + 2)$ in Fig. 5b. We kept the direct current (DC) component unchanged in this noise rejection process by subtracting it before applying the BPF and adding it after the inverse Fourier transform. As a result, we obtained the DKS shown in Fig. 6a. The oscillations in the H9 and H11 KER components around 0 and 280 fs are more clearly revealed in this figure than in Fig. 4a owing to the noise rejection by the BPF.

In the DKS for the perpendicular direction shown in Fig. 4b, we find that the oscillation in the H9 KER component is similar to that in the H3 KER component in the DKS for the parallel direction shown in Fig. 4a, although the signal of the H9 KER component is much lower than that of the H3 KER component. We can barely notice that the H11 and H13 KER components are also oscillating with a period similar to the vibrational period of ~16 fs, even though rapid fluctuations with a magnitude of ~10% of the averaged signal are contained in these components. This experimental result of the DKS for the perpendicular direction should be compared with the simulated DKS based on the theoretical model in which the H$_2^+$ ion dissociates via the $2p\pi_u$ state as shown in Fig. 3b.

We show the simulated DKS in Fig. 4d, which is in reasonable agreement with Fig. 4b in terms of the peak positions of the three KER components and the oscillatory behaviour around 280 fs. This similarity is also found in spectrograms in the frequency domain obtained by Fourier transforms of the experimental and simulated DKSs, which are shown in Fig. 5b,d, respectively.

We find that the (3,4) and (0,1) frequency components in the H13 KER component appearing in the simulated FKS in Fig. 5d do not appear in the experimental FKS in Fig. 5b. The difference-frequency peaks in the H11 KER component in Fig. 5b are somewhat scattered. These discrepancies might be caused by the disturbance from the intense parallel H3 and H5

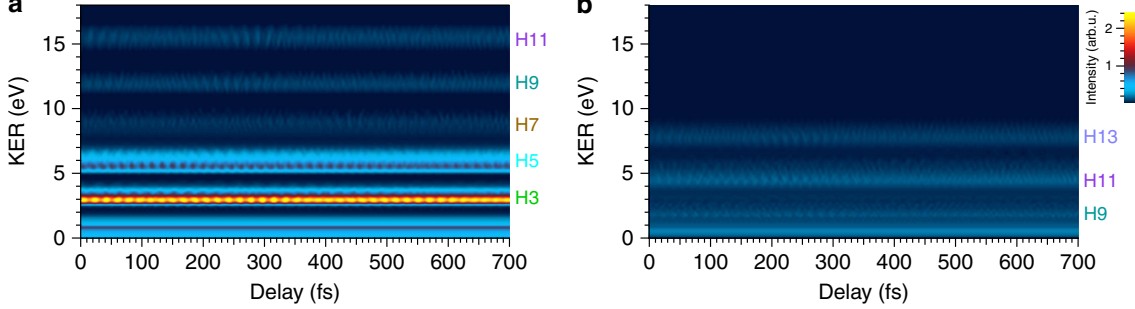

**Figure 5 | Magnitude squares of the Fourier transforms of the DKSs.** We referred to these spectrograms as frequency-KER spectrograms (FKSs). (**a**) FKS obtained by magnitude square of the Fourier transform of the DKS in Fig. 4a (parallel component). (**b**) FKS obtained by magnitude square of the Fourier transform of the DKS in Fig. 4b (perpendicular component). The logarithmic colour scales indicating the intensities in (**a**) and (**b**) are common. (**c**) FKS obtained by magnitude square of the Fourier transform of the DKS in Fig. 4c ($2p\sigma_u$). (**d**) FKS obtained by magnitude square of the Fourier transform of the DKS in Fig. 4d ($2p\pi_u$). The logarithmic colour scales indicating the intensities in (**c**) and (**d**) are common. The difference frequencies between the adjacent and next-adjacent vibrational states are depicted as grids with dotted lines and expressed in the parentheses as ($v$, $v+1$) and ($v$, $v+2$) on the top axis of each figure, where $v$ is the vibrational number. We show the number of each harmonic order contributing to each KER component on the right-hand side of each figure. The contours depicted with red dashed curves in (**a**) and (**b**) indicate the regions where the difference frequencies appear in (**c**) and (**d**), respectively. These regions are used as a BPF to reject the noise in the experimental DKSs in Fig. 4a,b.

**Figure 6 | Delay–KER spectrograms (DKSs) after applying the BPF.** (**a**) DKS of the parallel component obtained after applying the BPF depicted as red dashed contours in Fig. 5a to the DKS shown in Fig. 4a. (**b**) DKS of the perpendicular component obtained after applying the BPF depicted as red dashed contours in Fig. 5b to the DKS shown in Fig. 4b. The colour scales indicating the intensities in (**a**) and (**b**) are common and the same as those used in Fig. 4a,b.

KER components in the retrieval process for each sliced image using the pBasex method, or simply might be due to the low S/N ratio of the recorded data. In spite of these imperfections, the experimental FKS in Fig. 5b is well reproduced in the simulated

FKS in Fig. 5d, and hence we are assured that the perpendicular components of the $H^+$ fragment are created via the $2p\pi_u$ state.

On the basis of this assignment for the electronic state, we removed the frequency noise by applying the BPF, depicted as red

dashed contours in Fig. 5b, resulting in Fig. 6b. The DC component is kept unchanged in the BPF process. The oscillations around 280 fs are more clearly seen in all three KER components in this figure.

**Switching between two dissociation pathways**. The FKSs, shown in Fig. 5a,b, provide us clear evidence that the origin of the oscillation in the DKSs is the vibrational motion of the wavepacket created in the $1s\sigma_g$ state, while we cannot recognize from the positions and intensities of the distinct peaks in the FKSs how the relative phase of the oscillation in the time domain differs with the direction and KER component of the $H^+$ fragments. Therefore, we investigate specific parts of the DKSs shown in Figs 4a,b and 6a,b to demonstrate the importance of the timing of the oscillations.

We calculated the total $H^+$ yield in the H9 KER component for the perpendicular direction by integrating the DKS (Fig. 4b) in the KER region between 1.66 and 2.10 eV, which was divided by the square root of the peak KER so that the integration should be approximately proportional to the velocity. The resultant line profile is depicted as red crosses connected by dashed lines in Fig. 7a. The line profile obtained from the band-pass-filtered DKS (Fig. 6b) in the same manner is also shown as a red solid curve in the same figure. We denote these line profiles as $I_\perp^{H9}(\tau)$ and $\bar{I}_\perp^{H9}(\tau)$, respectively, where we define $\tau$ as the delay. The delay range of this figure is magnified around the time of the half-revival of the vibrational wavepacket because the wavepacket is spatially localized and the magnitude of the oscillation is maximized around this delay. The oscillation period is $\sim 17$ fs.

These traces of $I_\perp^{H9}(\tau)$ and $\bar{I}_\perp^{H9}(\tau)$ should be compared with the blue solid circles connected by dashed lines and the blue solid curve in Fig. 7a. We define these traces as $I_\parallel^{H9}(\tau)$ and $\bar{I}_\parallel^{H9}(\tau)$,

respectively. The trace of $I_\parallel^{H9}(\tau)$ is the total $H^+$ yield in the H9 KER component for the parallel direction (Fig. 4a) in the KER region between 11.36 and 12.24 eV, and the trace of $\bar{I}_\parallel^{H9}(\tau)$ is obtained from the band-pass-filtered DKS (Fig. 6a) in the same manner. The oscillation period is $\sim 16$ fs.

We observe that the peaks and valleys in the red curve are inverted in the blue curve or, equivalently, the phase of the oscillation in the red curve is shifted by $\pi$ in the blue curve. This characteristic is numerically evaluated using the distinction ratio, defined by $R_{\perp\parallel}(\tau) \equiv \left\{I_\perp^{H9}(\tau) - I_\parallel^{H9}(\tau)\right\} / \left\{I_\perp^{H9}(\tau) + I_\parallel^{H9}(\tau)\right\}$ $\left(\bar{R}_{\perp\parallel}(\tau) \equiv \left\{\bar{I}_\perp^{H9}(\tau) - \bar{I}_\parallel^{H9}(\tau)\right\} / \left\{\bar{I}_\perp^{H9}(\tau) + \bar{I}_\parallel^{H9}(\tau)\right\}\right)$, which is depicted as black solid triangles connected by dashed lines (black curve with shaded area) in Fig. 7a. The distinction ratio approaches its maximum value of 1 when $I_\perp^{H9}(\tau)$ or $\bar{I}_\perp^{H9}(\tau)$ is significantly larger than $I_\parallel^{H9}(\tau)$ or $\bar{I}_\parallel^{H9}(\tau)$, and it becomes close to $-1$ under the opposite condition. The distinction ratio should be 0 for the case of equal yields in both directions.

We find that the distinction ratio $\bar{R}_{\perp\parallel}(\tau)$ changes from a minimum of $0.16 \pm 0.063$ at $\tau = 271.5$ fs to a maximum of $0.34 \pm 0.066$ at $\tau = 279.8$ fs. In other words, the distinction ratio is switched from its minimum to its maximum by a change in the delay of the 9th-harmonic component in the probe APT of only 8 fs, even though the $H^+$ yields for the two pathways themselves are not reversed and the modulation depth of $0.18 \pm 0.091$ ($= 0.34-0.16$) is considerably smaller than the ideal value of 2.

We can explain how the dissociation pathways are selected by changing the delay of the probe APT using the schematic figures shown in Fig. 8a,b. At 8 fs before the time of the half-revival of the vibrational wavepacket, which is equal to 272.2 fs in our theoretical model, the vibrational wavepacket is confined around the inner turning point in the $1s\sigma_g$ potential, depicted as a thin

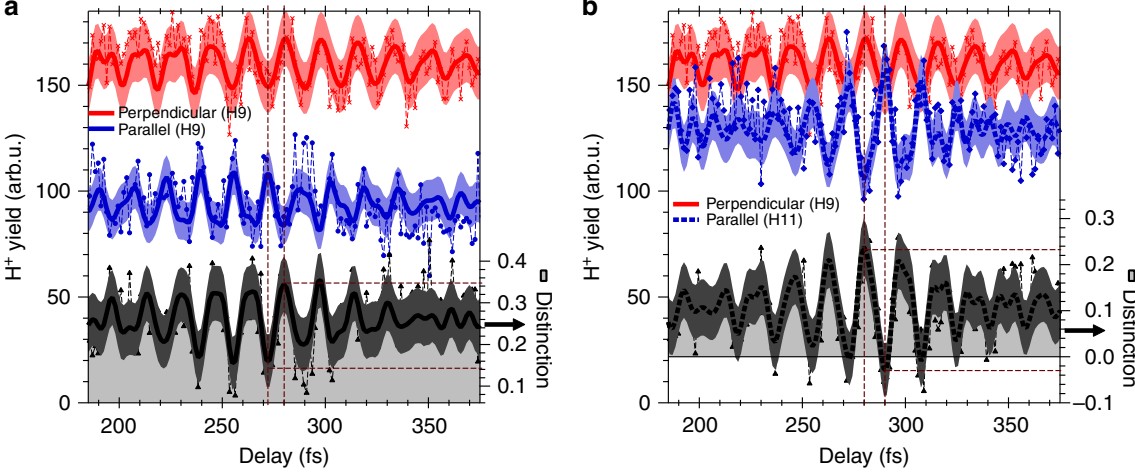

**Figure 7 | Line profiles of DKSs temporally magnified around the time of the half-revival of 280 fs.** (**a**) Line profile of H9 KER component in Fig. 4b (perpendicular, raw data) and that of Fig. 6b (perpendicular, band-pass-filtered) shown as red crosses connected by dashed lines and the red solid curve, respectively. Each trace is obtained by accumulating the KER component in the region between 1.66 and 2.10 eV in each DKS. The blue solid circles connected by dashed lines and the blue solid curve represent the line profile of the H9 KER component in Fig. 4a (parallel, raw data) and that of Fig. 6a (parallel, band-pass-filtered), respectively. Each trace is obtained by accumulating the KER component in the region between 11.36 and 12.24 eV in each DKS. Black solid triangles connected by dashed lines and the black solid curve with the shaded area represent the distinction ratio of the perpendicular component to the parallel component calculated from the above-mentioned raw traces and band-pass-filtered traces, respectively. The shaded area around each band-pass-filtered trace shows the error estimated from the $H^+$ yield signal counts. (**b**) Red crossings with connecting dashed lines and red solid curve are the same as those depicted in (**a**). The blue solid diamonds connected by dashed lines and the blue dotted curve represent the line profile of the H11 KER component in Fig. 4a (parallel, raw data) and that of Fig. 6a (parallel, band-pass-filtered), respectively. Each trace is obtained by accumulating the KER component in the region between 14.56 and 16.06 eV in each DKE. Black solid triangles connected by dashed lines and the black dotted curve with the shaded area represent the distinction ratio of the perpendicular component to the parallel component calculated from the above-mentioned raw traces and band-pass-filtered traces, respectively. The shaded area around each band-pass-filtered trace shows the error estimated from the statistics of the $H^+$ yield signal counts.

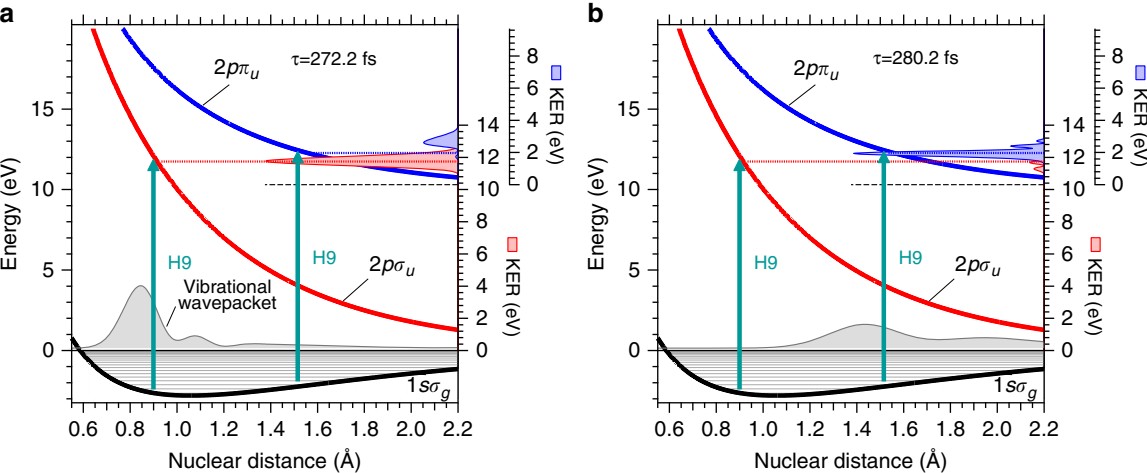

**Figure 8 | Schematic figures explaining how to switch between the two dissociation pathways.** The adiabatic potentials of the $1s\sigma_g$, $2p\sigma_u$ and $2p\pi_u$ states are depicted as black, red and blue solid curves, respectively. The KER spectrum of the $H^+$ fragment produced via the $2p\sigma_u$ state and that produced via the $2p\pi_u$ state are shown as thin red and blue curves with shaded areas on the right axis, respectively. The length of each vertical arrow accompanying the tag of H9 is proportional to the photon energy of the 9th-harmonic component. (**a**) The vibrational wavepacket arrives at the inner turning point 8 fs before the time of the half-revival of the wavepacket. The $1s\sigma_g$ state in the $H_2^+$ molecule is efficiently excited to the $2p\sigma_u$ state by the one-photon absorption of the 9th-harmonic component in the APT because the H9 photon energy approximately coincides with the energy gap between the $1s\sigma_g$ and $2p\sigma_u$ states at the inner turning point where the wavepacket is localized. (**b**) The vibrational wavepacket arrives at the outer turning point at the time of the half-revival of the wavepacket. The $1s\sigma_g$ state in the $H_2^+$ molecule is efficiently excited to the $2p\pi_u$ state by the one-photon absorption of the 9th-harmonic component in the APT because the H9 photon energy approximately coincides with the energy gap between the $1s\sigma_g$ and $2p\pi_u$ states at the outer turning point.

grey curve with a shaded area in Fig. 8a. The photoexcitation from the $1s\sigma_g$ state to the $2p\sigma_u$ state accompanied by one-photon absorption of the 9th harmonic component in the APT should be enhanced at this time because of the near coincidence of the photon energy with the energy difference between the $1s\sigma_g$ and $2p\sigma_u$ states at the inner turning point. In contrast, the excitation to the $2p\pi_u$ state is reduced owing to the fact that the magnitude of the wavepacket is considerably small near the outer turning point, where the excitation energy to the $2p\pi_u$ state is similar to the photon energy. After the half-period of the vibrational motion (8 fs), the wavepacket reaches the outer turning point, as shown in Fig. 8b, then the situation is reversed.

As a result, we can toggle the final product of $H^+ + H(1s)$ (via the $2p\sigma_u$ state) or $H^+ + H(2s)$ (via the $2p\pi_u$ state)[45] generated from the original reactant of $H_2$ by utilizing the time evolution of the vibrational wavepacket created in the intermediate product of $H_2^+$. Such a reaction control scheme utilizing the vibrational motion of an intermediate product was first proposed and theoretically investigated by Tannor *et al.* in a series of seminal papers[42,43], and then it was applied to experimental studies on reaction control in a sub-picosecond time scale, which is now part of the established scientific field of femtochemistry[39,40].

We note that the 11th-harmonic component efficiently excites the $1s\sigma_g$ state in the $H_2^+$ molecule to the $2p\sigma_u$ state at the inner turning point as well as the 9th-harmonic component. This is confirmed by calculating the line profiles of the H11 KER component in the DKS shown in Fig. 4a and that in Fig. 6a, which are depicted as blue diamonds connected by dashed lines and the blue dotted curve in Fig. 7b, respectively. When the $H^+$ yield in the H11 KER component for the parallel direction reaches its maximum at $\tau = 290.9$ fs, it exceeds the $H^+$ yield in the H9 KER component for the perpendicular direction, as shown in Fig. 7b, resulting in the reversal of $H^+$ yields. In fact, the distinction ratio obtained from the band-pass-filtered DKS, which is defined as $\bar{R}'_{\perp\parallel}(\tau) \equiv \left\{ \bar{I}_\perp^{H9}(\tau) - \bar{I}_\parallel^{H11}(\tau) \right\} / \left\{ \bar{I}_\perp^{H9}(\tau) + \bar{I}_\parallel^{H11}(\tau) \right\}$, is switched from the maximum value of $0.24 \pm 0.062$ at $\tau = 279.8$ fs to the

minimum value of $-0.03 \pm 0.057$ at $\tau = 289.5$ fs. The modulation depth is increased to $0.27 \pm 0.084$ ($= 0.24 - (-0.03)$) higher than that obtained for $\bar{R}_{\perp\parallel}(\tau)$, while the switching time is extended to 10 fs.

Note that the actual modulation depth of the distinction ratio (0.18 or 0.27) is far below the ideal value (2). This is owing to the fact that the magnitude of the oscillation of each KER component is much smaller than the magnitude of each DC background, which is considered to be due the dissociation process induced by the single pump or probe APT beam composed of multiple harmonic components ranging from H1 to H19. Hence, one possible origin of the DC background is a sequential ionization–excitation process that involves the absorption of two photons one by one in a single APT. Nevertheless, this kind of sequential process cannot convincingly explain why the excitation is induced far from the Franck–Condon region of $H_2$ to provide relatively low KER components. This is because the duration of the APT envelope is only $\sim 4$ fs, which is much shorter than the half-period of the vibrational motion of the wavepacket. Another possible origin may be the contribution of the doubly excited states of $H_2$, which are followed by the decay into the electronic states of the $H_2^+$ by autoionization[47,48]. The photon energies of the XUV harmonic components are, however, lower than most of the energies of the doubly excited states reported in the literature, and hence we need to assume two-photon excitation into the doubly excited states. The estimation of the probability of this autoionization process is beyond the scope of this paper.

## Discussion

We have demonstrated the control of the dissociation pathways of a $H_2^+$ molecule in a time scale of sub-10 fs using the vibrational dynamics in the $1s\sigma_g$ state of a $H_2^+$ molecular ion. The dissociation via the $2p\sigma_u$ state is enhanced by irradiating a probe pulse at the time when the vibrational wavepacket is localized near the inner turning point, while that via the $2p\pi_u$ state is reduced. This situation reverses upon changing the irradiation

time of the probe pulse by only 8 fs owing to wavepacket localization near the outer turning point. Note that this fastest reaction control of the simplest molecule cannot be realized without an intense 'a-few-pulse' APT in the XUV wavelength region on account of the high photon energies and the short pulse duration required for electronic excitations in the $H_2^+$ molecule.

The APT used in our experiment contains harmonic components that are unnecessary for the control and might cause a $H^+$ yield in the DC background. Therefore, we need a device to extract a single harmonic component from the APT (for example, the 13th-harmonic component for the pump pulse and the 9th-harmonic component for the probe pulse) with high throughput and without a dispersion, to improve the distinction ratio of the two pathways, even though such a useful device has not been realized.

By changing the viewpoint from the control of the reaction to the creation of multiple electronic states, we can claim that an attosecond electronic wavepacket composed of the $2p\sigma_u$ and $2p\pi_u$ states is generated upon the irradiation of the pump APT. This is because we can see in Fig. 3a,b that excitation to the $2p\sigma_u$ state with the 9th- and 11th-harmonic components and excitation to the $2p\pi_u$ state with the 13th-harmonic component should simultaneously occur in the Franck–Condon region after ionization. The 9th-, 11th- and 13th-harmonic components are

coherent with each other so as to form the APT, and hence the generated $2p\sigma_u$ and $2p\pi_u$ electronic states should also be coherent, resulting in the electronic wavepacket. The oscillating period of this wavepacket is estimated to be less than 1 fs from the energy difference between the two states in the Franck–Condon region. It may be possible to find a signature of the attosecond electronic wavepacket by scanning the delay of the probe APT with a sub-100-as delay step, because we observed an attosecond electronic wavepacket in a $N_2^+$ molecule in a similar experiment[41].

## Methods

**Experimental details.** The laser light source used to generate the XUV APT is a chirped pulse amplification system of Ti:sapphire laser delivering 14 fs pulses at a 100-Hz repetition rate. We inserted a pair of spectral filters fabricated with dielectric films into a regenerative amplifier in this laser system so as to compensate for the spectral narrowing during the amplification[49]. The maximum available energy of the pulse behind a grating pair compressor set in a vacuum chamber is 40 mJ.

After passing through a variable aperture, the laser pulse is focused with a concave mirror into a gas cell filled with a xenon gas target for HH generation. The focal length of the concave mirror (5 m) was designed to be as long as possible with the limitation of the size of our laboratory. The incident angle to the concave mirror is carefully adjusted such that the astigmatism of the pulse wavefront is removed. The gas pressure of the xenon and the pulse energy after optimizing the HH yield by changing the gas pressure and the diameter of the variable aperture are ~2 hPa and 15 mJ, respectively. The pulse energy of the HH pulse in the XUV

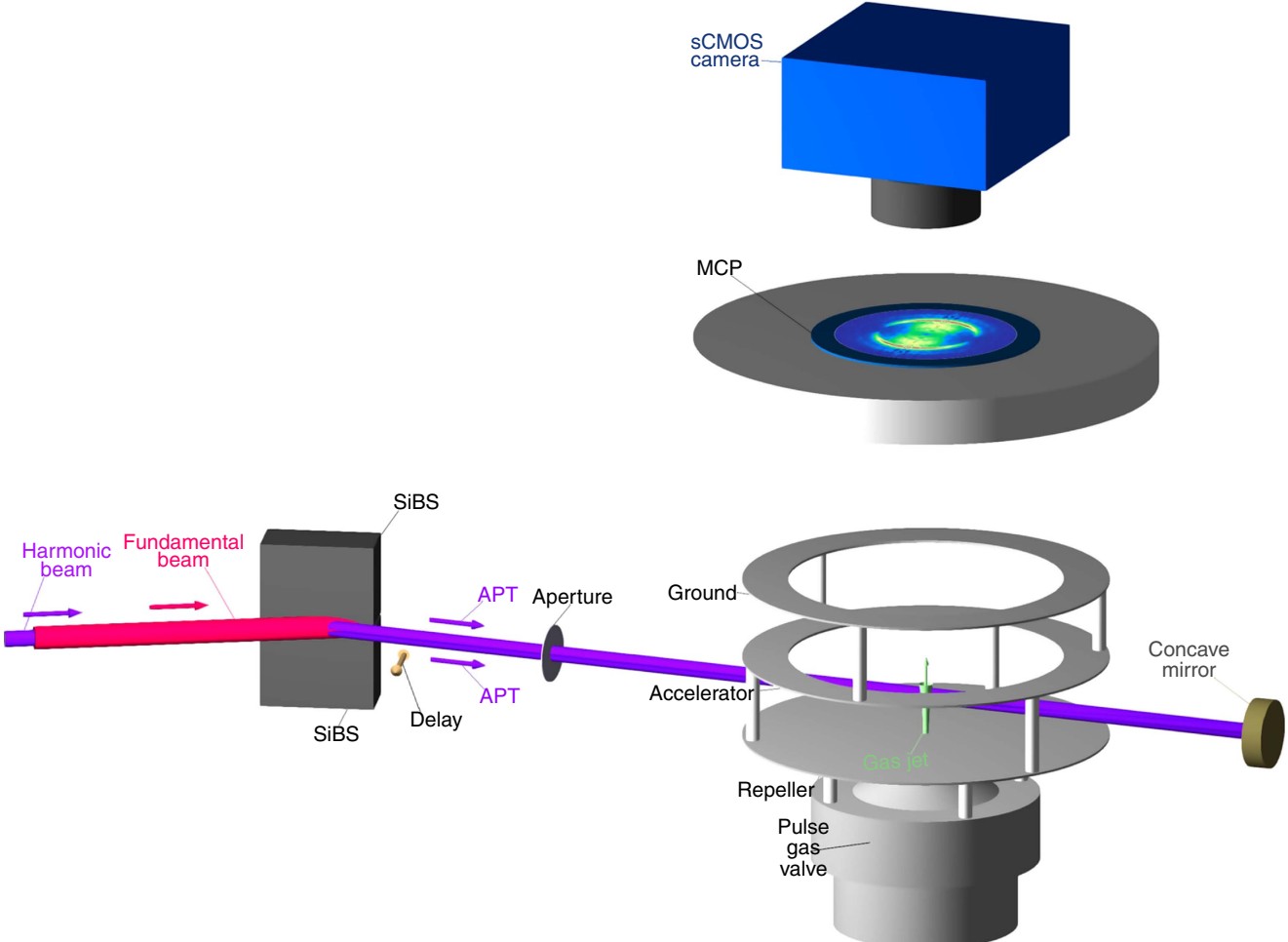

**Figure 9 | Schematic figure of experimental setup.** The harmonic beam, which forms an attosecond pulse train (APT) in the time domain, is introduced into a VMI ion spectrometer after reflection from a pair of silicon harmonic separator mirrors (SiBSs). The VMI ion spectrometer is composed of three electrodes (repeller, accelerator and ground), a micro-channel plate (MCP) and an sCMOS camera recording the fluorescent image of the phosphor screen attached to the MCP. The target $H_2$ molecules are injected through a pinhole at the centre of the repeller electrode as a gas jet ejected from a pulse gas valve. Two replicas of the APT propagating through an aperture are focused into the target gas jet using a concave mirror made of SiC.

region is expected to be much greater than 1 μJ because the conditions for HH generation are similar to those reported in refs 50,51.

The generated harmonic fields, the order of which ranges from the 3rd to 19th harmonics, co-propagate with the fundamental laser pulse in a 4-m long vacuum tube and are reflected by a pair of SiBSs to remove the intense fundamental laser pulse, as shown in Fig. 9. The harmonic fields are essentially phase locked to the fundamental laser field, and hence they form an APT in the time domain. We have estimated the duration of an APT envelope consisting of the XUV harmonic components (9th and higher) to be ∼4 fs by the interferometric autocorrelation measurement, as shown in Supplementary Fig. S3A in Supplementary Materials of ref. 41, and thus the number of attosecond pulses in the train envelope should be approximately three. We call this APT 'a-few-pulse APT'. The pulse durations of the 3rd- and 5th-harmonic components are both measured to be ∼7 fs, as demonstrated in Supplementary Fig. 5 in ref. 38. The temporal characteristic of the 7th-harmonic component is unknown. Nevertheless, detailed knowledge of the temporal characteristic is not required because the 7th-harmonic component does not play a significant role in the present study.

The APT is spatially split into two replicas by reflection near the boundary of the two SiBSs, one of which is mounted on a translation stage driven by a piezo actuator. We control the delay of one of the APT replicas by adjusting the position of the translation stage. After passing through an aperture with a diameter of 2 mm set behind the SiBS pair, the two replicas of the APT beam pass through a VMI ion spectrometer consisting of three electrodes, an MCP with a phosphor screen and a high-speed scientific CMOS (sCMOS) camera to record a fluorescent image of ions appearing on the phosphor screen. The target $H_2$ molecules are supplied through a pinhole drilled at the centre of the repeller electrode as a gas jet ejected from a pulse gas valve attached to but insulated from the repeller electrode. The position of the pinhole or, equivalently, the centre position of the VMI ion spectrometer is set ∼5 mm aside from the path of the APT beams so as to minimize the ion signal originating form the one-photon absorption of the unfocused APT beams.

The APT beams are reflected back and focused into the target gas jet using a concave mirror made of crystalline silicon carbide (SiC) with a radius of curvature of 200 mm, which has been newly fabricated for the current experiment. The ion signal is considerably enhanced by this crystalline SiC mirror compared with that by the amorphous SiC mirror used in previous works[36,38,52], although we do not quantitatively estimate the absolute ion yields. We suppose that the reflectivity of the crystalline SiC mirror is somewhat higher than that of the amorphous SiC mirror, and this might explain why we can observe ion signals that could not be found in previous works. We show the calculated reflectivity of the SiC mirror in Supplementary Fig. 3. We do not exactly specify the distribution of the harmonic components contained in the APT beam at the focal point, while we expect that it will be similar to those exhibited in Supplementary Fig. 1 in ref. 38 and Fig. S2B in the Supplementary Materials of ref. 41.

The ion fragments of $H^+$ are accelerated by the three electrodes in the VMI ion spectrometer and collide with the MCP. The voltages applied to the electrodes are adjusted so that the position of the detected ions can indicate their velocity, and the voltages are typically 2,000 V for the repeller electrode, 1,690 V for the accelerator electrode and 0 V for the ground electrode. The sensitivity of the MCP is temporarily gated by applying a pulsed high voltage with a pulse duration of 100 ns at the flight time of $H^+$ fragments to eliminate the signals from other ion fragments. We can determine the position of each ion hitting the MCP by observing each fluorescent point on the phosphor screen attached to the MCP. The image of each fluorescent point is recorded by an sCMOS camera.

Note that we have completely eliminated the huge signal of the parent ion, $H_2^+$, which is mainly generated by the one-photon absorption of the XUV components in the APT and does not exhibit the periodical modulations due to delay scanning between the two APT beams, by applying a square gate voltage to the MCP with a magnitude of 2,500 V without a DC off-set voltage. This is an improvement of our experimental setup compared with that used in our previous experiment[36].

We scanned the delay of one of the APT beams by translating the piezo stage with steps of 790 nm, which should be identical to steps in the delay of ∼1.4 fs, considering that the incident angle to the SiBS is set to 75°. The number of delay points is 530, which ensures that the range of the scanning delay is more than 700 fs. We acquired 250 fluorescent images in 5 s with an sCMOS camera synchronously operated with laser shots at every delay step, with the scanning of the delay repeated six times. Hence, one recorded image contains the information of the ion yields from two laser shots because the repetition rate of the laser pulse is 100 Hz and the image of the $H^+$ fragment at each delay is composed of the ion-counting data of 1,500 fluorescent images obtained from 3,000 laser shots.

**Retrieval of the sliced image of $H^+$ fragments.** The pBasex[44] Abel transform method for image inversion is applied to the image in Fig. 1a, to obtain the sliced image in the left panel of Fig. 1b. In this method, we can choose even-order Legendre polynomials as basis functions for the polar angle coordinate to expand the sliced image because the target image shown in Fig. 1a,b has been symmetrized in both the x- and z-directions. Nevertheless, it is a non-trivial issue how many orders of Legendre polynomials are needed to retrieve an appropriate sliced image, even though we can expect that the 4th order will be sufficient owing to the fact that the $H^+$ fragments are generated by a two-photon process. We have applied the expansion up to the 6th order and confirmed that the sliced image retrieved

using expansion coefficients up to the 4th order is in reasonable agreement with that retrieved using another method developed by Vrakking[53]. Therefore, we have used the expansion coefficients up to the 4th order to retrieve the sliced images.

**Theoretical model for DKSs.** In the theoretical model for calculating DKSs, we assume that the initial vibrational wavepacket in the $1s\sigma_g$-state $H_2^+$ molecule, $\psi_{g0}(R) = \sum_v a_v \chi_v^g(R)$, is generated by the irradiation of the preceding APT, and that the following APT irradiated at the delay time $\tau$ excites the $H_2^+$ molecule to the $2p\sigma_u$ and $2p\pi_u$ states, where we denote the $v$th vibrational wavefunction in the $1s\sigma_g$ state at the internuclear distance $R$ as $\chi_v^g(R)$. The eigenenergy of $\chi_v^g(R)$ is defined as $\hbar\omega_v^g$, where $\hbar$ is Plancks constant divided by $\pi$. The complex amplitude of the $v$th vibrational wavefunction $a_v$ is calculated from the ionization model proposed in ref. 38, as shown in Supplementary Fig. 4.

By applying conventional time-dependent perturbation theory to the Schrödinger equation, we find that the amplitude of the first-order transition to the $2p\sigma_u$ state is proportional to $T^{\sigma_u}(\omega^{\sigma_u};\tau)$, defined as

$$T^{\sigma_u}(\omega^{\sigma_u};\tau) \equiv \sum_v \mathcal{M}^{\sigma_u}(\omega^{\sigma_u};\omega_v^g)\tilde{G}(\omega^{\sigma_u}-\omega_v^g)a_v e^{-i\omega_v^g\tau}, \quad (1)$$

where the dipole transition matrix elements are given by

$$\mathcal{M}^{\sigma_u}(\omega^{\sigma_u};\omega_v^g) \equiv \int_0^\infty dR \chi^{\sigma_u}(\omega^{\sigma_u};R)\mu^{\sigma_u}(R)\chi_v^g(R). \quad (2)$$

We define the nuclear wavefunction in the $2p\sigma_u$ state with a continuum eigenenergy of $\hbar\omega^{\sigma_u}$ as $\chi^{\sigma_u}(\omega^{\sigma_u};R)$ and the transition dipole from the $1s\sigma_g$ state to the $2p\sigma_u$ state as $\mu^{\sigma_u}(R)$. The eigenenergy of $\hbar\omega^{\sigma_u}$ is equivalent to the KER because we measure the energy from the dissociation limit common to the $1s\sigma_g$ and $2p\sigma_u$ states. The positive-frequency part of the Fourier transform of the APT field at the angular frequency of $\Omega$ is written as $\tilde{G}(\Omega)$. The DKS is obtained as $|T^{\sigma_u}(\omega^{\sigma_u};\tau)|^2$.

The transition amplitude to the $2p\pi_u$ state $T^{\pi_u}(\omega^{\pi_u};\tau)$ is given by simply replacing $\sigma_u$ with $\pi_u$ in equations (1) and (2). Nevertheless, it should be noted that the eigenenergy of the $2p\pi_u$ state $\hbar\omega^{\pi_u}$ is composed of the energy of the dissociation limit of the $2p\pi_u$ state $\hbar\omega_{\text{diss}}^{\pi_u}$ measured from the dissociation limit of the $1s\sigma_g$ and $2p\sigma_u$ states and the KER $\hbar\omega_{\text{KER}}$, namely $\hbar\omega^{\pi_u} = \hbar\omega_{\text{KER}} + \hbar\omega_{\text{diss}}^{\pi_u}$.

**Data availability.** The data that support the findings of this study are available from the corresponding author upon request.

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

## Acknowledgements

We thank Professor Shuntaro Watanabe (Tokyo University of Science) and Professor Jiro Itatani (University of Tokyo) for providing us with the technical information concerning a piezo gas valve used to supply target molecules to the VMI chamber. This work was part of the Advanced Photon Science Alliance (APSA) research project, commissioned by MEXT of Japan, and has contributed to the missions of CREST and PRESTO studies commissioned by JST. Y.N., T.O., E.J.T. and K.M. gratefully acknowledge financial support from Grants-in-Aid for Scientific Research Nos. 26247068, 26600123, 25286074, 26600122 and 26220606.

## Author contributions

Y.N. developed the laser system, conducted the experiment, partially contributed to the data analysis and wrote the manuscript. Y.F. was responsible for the data acquisition and analysis. T.O. designed and built the VMI spectrometer and developed the data acquisition software. A.A.E. was also involved in the development of the laser system. E.J.T. developed the HH beam line and XUV spectrograph. K.Y. supervised the experiment on VMI. K.M. directed the research in accordance with the Extreme Photonics research project of RIKEN and the APSA research project of MEXT.

## Additional information

**Competing financial interests:** The authors declare no competing financial interests.

