## [Peer Review File · Nature Communications]

Reviewer #1 (Remarks to the Author):

The authors claim to have demonstrated that a time evolution of the vibrational wavepacket can be used to control the dissociation pathway through selection of excitation to either the $2p\sigma_u$ electronic state or the $2p\pi_u$ electronic state in the hydrogen molecular ion. Both excited states are created by coherent excitation from the $1s\sigma_g$ ground electronic state of H_2^+ , using one-photon absorption of an attosecond pulse train with a train-envelope duration of ~ 4 fs. The time-dependent oscillation in the ion yield from the $2p\sigma_u$ state is complementary to that from the $2p\pi_u$ state, at delays approaching the time of the half revival of the wavepacket. The switching time from one dissociation channel to the other is around 8 fs, which the authors claim to be the fastest control of a molecular reaction in the simplest molecule on the basis of the Tannor-Kosloff-Rice scheme.

Although the authors do point out that there have been other demonstrations of the Tannor-Kosloff-Rice control scheme in the literature, the fact that the experiments are performed on the simplest molecule, H_2^+ , and demonstrate control on an ~ 8 fs timescale, in my mind does set this work apart from that published previously. In my view the work is sufficiently novel to warrant publication in Nature Communications. I also believe that the paper will be of general interest to readers: I recommend publication in Nature Communications.

I believe the data provided by the authors are convincing, and they go to great lengths to explain their experimental findings. One wonders in fact whether the paper would benefit from being somewhat more concise, leaving some of the detail explanations to supplementary material.

Some of the oscillations in the experimental time-delay scans shown in Figure 5 are quite hard to see, particularly those associated with the higher harmonics. The subsequent Fourier transform analysis shown in Figure 6, and the filtered time-delay data shown in Figures 7 and 8 are, however, convincing. The analysis of the data looks robust to me, although perhaps the errors on the distinction ratios could do with defining towards the end of the manuscript.

In my view, the authors have provided a beautiful demonstration of coherent control in the simplest molecule. For that reason I believe the experiments do capture the imagination, and have the potential to influence thinking in the field, even though conceptually, it could be argued that the experiments are not completely new.

Reviewer #2 (Remarks to the Author):

This manuscript reports on a beautiful XUV pump - XUV probe experiment performed with an attosecond pulse train with a few-femtosecond envelope to measure a vibrational wave-packet in H_2^+ . This vibrational wave packet is exploited to demonstrate control over the dissociation of H_2^+ into either $H(1s)$ and H^+ or $H(2s)$ and H^+ . Vibrational wave packet motion with a period of 8 fs is observed and wave-packet control of molecular dissociation on the shortest ever-achieved time scale is demonstrated.

These results constitute a clear and impressive advance in the field of attosecond science. I enthusiastically recommend publication in Nature Communications. I only have a few remarks that I would like to bring to the attention of the authors:

- 1) Neither the title nor the abstract emphasize the XUV pump - XUV probe aspect of the present work, which I perceive as the most important. I propose that the authors make changes in this sense.
- 2) Do the authors observe modulations in the H_2^+ signal? Why / not? This could be interesting to

discuss.

3) What is the estimated reflectivity of the SiC mirror in the relevant range of photon energies?

4) A short qualitative explanation of theory would be very helpful. In the present version of the manuscript, it is impossible to judge the quality of the theoretical work without reading the references. One or two sentences mentioning the methods and approximations used would be very helpful and informative.

5) Can states of H₂⁺ lying above the 2pπ_u state be neglected from energy considerations or is it an assumption made to simplify the calculations?

6) On p. 10, col. 2, the subscripts of 1σ_g should be g, not p

Reviewer #3 (Remarks to the Author):

Review NCOMMS-16-06252, Nabekawa et al.

The authors report an interesting XUV-pump, XUV-probe experiment in H₂. A relatively high pulse energy laser system is used to generate sufficient high harmonic flux to both ionize H₂ to H₂⁺, and then dissociate some variable time delay later. This experiment is a significant achievement and the authors are to be congratulated on their efforts, particularly in light of being able to generate significant ion signal so as to operate a velocity-map imaging spectrometer.

With that said, unfortunately I do not think this work warrants publication in Nature Communications. While the experimental achievements will be of interest to scientists in the field of ultrafast molecular dynamics, I do not think there is much scope for wider interest. The reason behind this statement is near-IR pump-probe and IR-pump, XUV-probe (and vice versa) observations of this prototypical system have reached many similar conclusions, in that vibrational and electronic wavepacket simulations accurately describe the ultrafast dynamics, even down to few femtoseconds.

There are a number of points which will significantly improve this manuscript should the authors seek to publish elsewhere (which they are strongly encouraged to do so).

1. There is no mention of the distribution of vibrational states generated in the 1σ_g state of H₂⁺. This is a required input to the simulation hence the authors are advised to give details. How does this distribution depend on the spectrum of the APT?

2. The manuscript would benefit from a detailed potential energy diagram early on so as to facilitate discussion of the ionization and dissociation routes.

3. There is no discussion of the conversion of pixel to velocity or KER, nor any discussion of uncertainties in this conversion. There is no perfect calibration of a VMI instrument, as the resolution changes as a function of KER/velocity, and no mention is made here. Also, Fig 2(a), labelling as "ch" is insufficient.

4. Again, Fig2(a), replace the term "intensity" with "signal (arb. units) or similar. Discussing intensity in the text is misleading.

5. Page 5, Para 2 "The KER peaks....". Make a direct comparison between the KER peaks in Fig 3 with those shown in Fig 4.

6. Page 8, para "In the DKS for the...", mention SNR without quantifying. Either quantify or find another more rigorous way of describing this.

7. There are no error bars on Fig 8, which would be a great improvement.

8. The invocation of the Tannor-Kosloff-Rice scheme is both unnecessary and a little misleading. While the dissociation of a diatomic is a "molecular reaction", it is hardly needed. This work does not exhibit any quantum control, i.e there is no specification of a target outcome, hence using the language of this field is a little excessive. Manipulating vibrational and electronic wavepackets is sufficiently interesting to the specialist.

Reply to reviewers
Manuscript No. NCOMMS-16-06252, entitled
“Sub-10-fs control of dissociation pathways in hydrogen
molecular ion with a-few-pulse attosecond pulse train”

We are grateful to all the reviewers for their useful suggestions to improve the quality of our paper.

In the following, we give point-by-point responses to the individual comments of the reviewers and describe the specific changes and additions that we have made to the manuscript. The changes in the main text (Sub10fsDissCntrl_H2_rvsd_withChangeMarks.pdf) are written in blue-green characters and we have newly added Supplementary Information including four Supplementary Figures to respond to the reviewers' comments.

Reply to Reviewers

Reply to reviewer #1

We thank the reviewer for his/her positive evaluation of our manuscript. We have addressed the following issues raised by the reviewer.

(a) I believe the data provided by the authors are convincing, and they go to great lengths to explain their experimental findings. One wonders in fact whether the paper would benefit from being somewhat more concise, leaving some of the detail explanations to supplementary material.

We acknowledge the suggestion from the reviewer to reduce the length of the manuscript, even though we decided not to move part of the contents to supplementary information. The content type for original research in *Nature Communications* is an Article, which may range in length from a short communication to a more in-depth study (please refer to URL: http://www.nature.com/ncomms/authors/content_types.html), and we intended to write our manuscript to describe an in-depth study. Therefore, we suppose that it would be better to show the research contents in the main text in as much detail as possible. We can move part of EXPERIMENT to Supplementary Information or Method, if the manuscript length exceeds the limit of *Nature Communications*.

(b) Some of the oscillations in the experimental time-delay scans shown in Figure 5 are quite hard to see, particularly those associated with the higher harmonics. The subsequent Fourier transform analysis shown in Figure 6, and the filtered time-delay data shown in Figures 7 and 8 are, however, convincing. The analysis of the data looks robust to me, although perhaps the errors on the distinction ratios could do with defining towards the end of the manuscript.

We have added a shaded area to every band-pass-filtered trace in Figs. 8(a) and (b) to illustrate the errors estimated from the signal counts. We describe these shaded areas in the last phrase of each caption as

The shaded area around each band-pass-filtered trace shows the error estimated from the H^+ yield signal counts.

We have also added the error range to the distinction ratios. Please refer to the right columns of page 10 and page 11.

Reply to reviewer #2

We thank the reviewer very much for his/her strong recommendation for the publication of our manuscript in *Nature Communications*. We have addressed all the issues pointed out by the reviewer and corrected typographical errors in the revised manuscript as follows.

1) Neither the title nor the abstract emphasize the XUV pump - XUV probe aspect of the present work, which I perceive as the most important. I propose that the authors make changes in this sense.

We gratefully acknowledge the reviewer's suggestion. We have made some corrections in the abstract. First, we added the phrase "in the extreme ultraviolet (XUV) wavelength region" after "an attosecond pulse train" to clarify the abbreviation "XUV". Then, we changed the last sentence from

This result can be classified as the fastest control of a molecular reaction in the simplest molecule on the basis of the Tannor–Kosloff–Rice scheme.

to

This result can be classified as the fastest control of a molecular reaction in the simplest molecule on the basis of the XUV-pump and XUV-probe scheme.

We removed the statement on the classification into the Tannor–Kosloff–Rice scheme from the abstract because of the criticism from reviewer #3 that such references were unnecessary. Instead, we emphasize the difference between the final products of the $2p\sigma_u$ and $2p\pi_u$ states in the last paragraph in INTRODUCTION and refer to the Tannor–Kosloff–Rice papers to clarify who originally proposed this kind of control scheme, as follows.

The final product of the $2p\sigma_u$ state is $H^+ + H(1s)$, while that of the $2p\pi_u$ state is $H^+ + H(2s)$. Hence, our experimental result can be regarded as the simplest reaction control in accordance with the method proposed by Tannor, Kosloff, and Rice[42,43].

2) Do the authors observe modulations in the H_2^+ signal? Why / not? This could be interesting to discuss.

In principle, two photons (one for each APT beam) or more are needed to modulate an ion signal, while most of the parent ions, H_2^+ , are generated via one-photon absorption (the sequential process of excitation to and de-excitation from the repulsive states should be negligible). In practice, the signal of the parent ion is sufficiently large to saturate the MCP detector in our VMI spectrometer. Hence, we did not perform delay scanning for the H_2^+ signal.

We briefly refer to the parent ion in EXPERIMENT (right column of page 3 in the revised manuscript) as follows.

Note that we have completely eliminated the huge signal of the parent ion, H_2^+ , which is mainly generated by the one-photon absorption of the XUV components in the APT and does not exhibit the periodical modulations due to delay scanning between the two APT beams, by applying a square gate voltage to the MCP with a magnitude of 2500 V without a DC offset voltage. This is an improvement of our experimental setup compared with that used in our previous experiment[36].

3) What is the estimated reflectivity of the SiC mirror in the relevant range of photon energies?

We show the reflectivity calculated from the known optical constant in Supplementary Figure 1 and refer to this figure in the left column of page 3 as follows.

We show the calculated reflectivity of the SiC mirror in Supplementary Figure. 1.

The reflectivity in the relevant range of photon energies is estimated to be 30~40% in this figure.

4) A short qualitative explanation of theory would be very helpful. In the present version of the manuscript, it is impossible to judge the quality of the theoretical work without reading the references. One or two sentences mentioning the methods and approximations used would be very helpful and informative.

We added explanations of our theoretical model used to calculate DKSs as follows.

In this model, we assume that the initial vibrational wavepacket in the $1s\sigma_g$ -state H_2^+ molecule, $\psi_{g0}(R) = \sum_{\nu} a_{\nu} \chi_{\nu}^g(R)$, is generated by the irradiation of the preceding APT, and that the following APT irradiated at the delay time τ excites the H_2^+ molecule to the $2p\sigma_u$ and $2p\pi_u$ states, where we denote the ν th vibrational wavefunction in the $1s\sigma_g$ state at the internuclear distance R as $\chi_{\nu}^g(R)$. The eigenenergy of $\chi_{\nu}^g(R)$ is defined as $\hbar\omega_{\nu}^g$, where \hbar is Planck's constant divided by π . The complex amplitude of the ν th vibrational wavefunction a_{ν} is calculated from the ionization model proposed in ref.[38], as shown in Supplementary Figure 4. By applying conventional time-dependent perturbation theory to the Schrödinger equation, we find that the amplitude of the first-order transition to the $2p\sigma_u$ state is proportional to $T^{\sigma_u}(\omega^{\sigma_u}; \tau)$, defined as

$$T^{\sigma_u}(\omega^{\sigma_u}; \tau) \equiv \sum_{\nu} \mathcal{M}^{\sigma_u}(\omega^{\sigma_u}; \omega_{\nu}^g) \tilde{G}(\omega^{\sigma_u} - \omega_{\nu}^g) a_{\nu} e^{-i\omega_{\nu}^g \tau}, \quad (\text{R1})$$

where the dipole transition matrix elements are given by

$$\mathcal{M}^{\sigma_u}(\omega^{\sigma_u}; \omega_{\nu}^g) \equiv \int_0^{\infty} dR \chi^{\sigma_u}(\omega^{\sigma_u}; R) \mu^{\sigma_u}(R) \chi_{\nu}^g(R). \quad (\text{R2})$$

We define the nuclear wavefunction in the $2p\sigma_u$ state with a continuum eigenenergy of $\hbar\omega^{\sigma_u}$ as $\chi^{\sigma_u}(\omega^{\sigma_u}; R)$ and the transition dipole from the $1s\sigma_g$ state to the $2p\sigma_u$ state as $\mu^{\sigma_u}(R)$. The eigenenergy of $\hbar\omega^{\sigma_u}$ is equivalent to the KER because we measure the energy from the dissociation limit common to the $1s\sigma_g$ and $2p\sigma_u$ states. The positive-frequency part of the Fourier transform of the APT field at the angular frequency of Ω is written as $\tilde{G}(\Omega)$. The DKS is obtained as $|T^{\sigma_u}(\omega^{\sigma_u}; \tau)|^2$. In this calculation, the magnitude of each harmonic component in the APT is adjusted...

(right column of page 7~left column of page 8)

and

The transition amplitude to the $2p\pi_u$ state $T^{\pi_u}(\omega^{\pi_u}; \tau)$ is given by simply replacing σ_u with π_u in Eqs (1) and (2). Nevertheless, it should be noted the eigenenergy of the $2p\pi_u$ state $\hbar\omega^{\pi_u}$ is composed of the energy of the dissociation limit of the $2p\pi_u$ state $\hbar\omega_{\text{diss}}^{\pi_u}$ measured from the dissociation limit of the $1s\sigma_g$ and $2p\sigma_u$ states and the KER $\hbar\omega_{\text{KER}}$, namely $\hbar\omega^{\pi_u} = \hbar\omega_{\text{KER}} + \hbar\omega_{\text{diss}}^{\pi_u}$.

(left column of page 9)

5) Can states of H_2^+ lying above the $2p\pi_u$ state be neglected from energy considerations or is it an assumption made to simplify the calculations?

We could not exactly assign the H^+ fragments originating from highly excited states lying above the $2p\pi_u$ state in the present experimental data, although we may have found the signatures of the contribution from such highly excited states in a separate experiment recently performed. We have addressed this issue in the left column of page 6 as follows.

We neglect the excited states other than the $2p\sigma_u$ and $2p\pi_u$ states in our analysis for simplicity, although the measured KER spectrograms may contain minor contributions from highly excited states that are not taken into account.

6) On p. 10, col. 2, the subscripts of $1s\sigma$ should be g, not p.

Thank you very much for pointing out this typographical error. We have corrected the subscripts from p to g .

Reply to reviewer #3

We thank the reviewer for his/her beneficial suggestions to improve the quality of our manuscript. In the following, we hope to convince the reviewer of the importance of our study.

^(a)I do not think there is much scope for wider interest. The reason behind this statement is near-IR pump-probe and IR-pump, XUV-probe (and vice versa) observations of this prototypical system have reached many similar conclusions, in that vibrational and electronic wavepacket simulations accurately describe the ultrafast dynamics, even down to few femtoseconds.

The most important point in our study is the switching of the final products after the irradiation of the second XUV attosecond pulse train (APT) by changing the delay time. The neutral H atom produced after the dissociation should be the ground electronic state with a delay time for 272.2 fs, while it should be the excited state (H(2s)) for a delay time of 280.2 fs. We cannot find, to the best of our knowledge, any similar conclusions to this result.

Our new finding is owing to the fact that the photon energy of H9 or higher in the XUV region is sufficiently high to excite the $1s\sigma_g$ state to the $2p\pi_u$ state at the outer turning point of the vibrational wavepacket by absorbing one-photon, and also to excite the $1s\sigma_g$ state to the $2p\sigma_u$ state at the inner turning point. This scheme could have not been experimentally realized without our high-flux, sub-10-fs, XUV APT source. In this sense, another important or novel feature of this study is the demonstration of the XUV-pump and XUV-probe experiment itself. This novelty is proven by the fact that the reviewer does not cite the XUV-pump and XUV-probe experiment as an example of an observation scheme.

As the reviewer points out, we acknowledge that theoretical simulations can describe ultrafast dynamics. Our study reported in this manuscript, however, is not only a simple verification of the theory. We have introduced a novel technology, in addition to fundamental science, for the molecular reaction by utilizing the ultrafast dynamics induced and controlled with XUV APTs. Therefore, we are convinced that our study will attract wide interest from both fundamental scientists in molecular physics and applied scientists in ultrafast laser technology, and that the manuscript is suitable for publication in *Nature Communications*.

1) There is no mention of the distribution of vibrational states generated in the $1s\sigma_g$ state of H_2^+ . This is a required input to the simulation hence the authors are advised to give details. How does this distribution depend on the spectrum of the APT?

We have shown the amplitude of each vibrational state in the initial vibrational wavepacket in Supplementary Figure 4. We briefly explain how we obtained the amplitude in the figure caption. Please refer to Supplementary Information. We cite this Supplementary Figure in the left column of page 8.

2) The manuscript would benefit from a detailed potential energy diagram early on so as to facilitate discussion of the ionization and dissociation routes.

Thank you for the useful suggestion. However, the manuscript format recommended by *Nature Communications* consists of *Introduction, Results, Discussion, and Methods* in this order. We suppose that the potential energy diagram should be included in *Discussion* written after *Results*, even though we do not exactly follow the recommended format. Therefore, the location of the potential energy diagram has not been changed, and we consider that this diagram is sufficiently detailed for the purpose of this study.

3) There is no discussion of the conversion of pixel to velocity or KER, nor any discussion of uncertainties in this conversion. There is no perfect calibration of a VMI instrument, as the resolution changes as a function of KER/velocity, and no mention is made here. Also, Fig 2(a), labelling as "ch" is insufficient.

We have shown the calibration curve used to relate the pixel number to the KER in Supplementary Figure 2, and we explain how we determined the relation between the pixel number and the KER in the caption. The resultant accuracy and resolution are adequate for our analysis in the main text. Please refer to Supplementary Information. We cite this Supplementary Figure in the right column of page 4 as follows.

Note that we calibrated the kinetic energy release (KER), which is twice the KE, in accordance with the manner described in Supplementary Note 3 in ref.[38]. The relation between the pixel number and the KER is depicted in Supplementary Figure 2.

We have changed the labels on both the horizontal and vertical axes in Fig. 2(a) from "ch" to "pixel", and clarify the pixel being referred to in the caption as follows.

(a) Image of H^+ fragment obtained from the accumulation of all the acquired data. We extracted each image with a size of 595 pixels \times 595 pixels from each image data recorded by the sCMOS camera such that the center of the fragment image is located at pixel [297, 297], where the pixel number starts from 0 in the extracted image.

We have also added an explanation of the image detection system of the VMI in the right column of page 3 as follows.

We can determine the position of each ion hitting the MCP by observing each fluorescent point on the phosphor screen attached to the MCP. The image of each fluorescent point is recorded by an sCMOS camera.

4) Again, Fig2(a), replace the term “intensity” with “signal (arb. units)” or similar. Discussing intensity in the text is misleading.

We have replaced “intensity” with “signal” in Fig. 2(a). The word “intensity” in the caption and the main text has been corrected so as not to mislead readers.

5) Page 5, Para 2 “The KER peaks...”. Make a direct comparison between the KER peaks in Fig 3 with those shown in Fig 4.

We show a direct comparison between the experimental and theoretical KER profiles in Supplementary Figures 3(a) and (b). Please refer to Supplementary Information. We cite these Supplementary Figures in the left columns of page 5 and page 6, respectively.

6) Page 8, para “In the DKS for the...”, mention SNR without quantifying. Either quantify or find another more rigorous way of describing this.

We have explained what disturbs the relevant modulation of the signal in a more specific way in the left column of page 9 as follows.

We can barely notice that the H11 and H13 KER components are also oscillating with a period similar to the vibrational period of ~ 16 fs, even though rapid fluctuations with a magnitude of approximately 10% of the averaged signal are contained in these components.

7) There are no error bars on Fig 8, which would be a great improvement.

We have indicated the error of each trace in Figs. 8(a) and (b) as a shaded area and added a sentence to explain the shaded area in the caption as follows.

The shaded area around each band-pass-filtered trace shows the error estimated from the H^+ yield signal counts.

8) The invocation of the Tannor-Kosloff-Rice scheme is both unnecessary and a little misleading. While the dissociation of a diatomic is a “molecular reaction”, it is hardly needed. This work does not exhibit any quantum control, i.e there is no specification of a target outcome, hence using the language of this field is a little excessive. Manipulating vibrational and electronic wavepackets is sufficiently interesting to the specialist.

To argue against the claim of the reviewer, we specify the two different target outcomes, one of which is ($H^+ + H(1s)$) via the $2p\sigma_u$ state, and the other is ($H^+ + H(2s)$) via the $2p\pi_u$ state. We believe that the quantum state of $H(1s)$ differs from that of $H(2s)$, and thus we use the phrase “reaction control” to emphasize the difference of the two products. In order to clarify this fact, we added the sentences in the left column of page 2.

The final product of the $2p\sigma_u$ state is $H^+ + H(1s)$, while that of the $2p\pi_u$ state is $H^+ + H(2s)$. Hence, our experimental result can be regarded as the simplest reaction control in accordance with the method proposed by Tannor, Kosloff, and Rice[42,43].

In spite of the above claim for the reaction control, we acknowledge that the repeated references to “the Tannor-Kosloff-Rice scheme” are unnecessary. We have omitted this phrase from the abstract and SUMMARY AND PROSPECTS. Instead, the last sentence of the abstract has been changed to

This result can be classified as the fastest control of a molecular reaction in the simplest molecule on the basis of the XUV-pump and XUV-probe scheme.

in accordance with the suggestion from reviewer #2. The first sentence in SUMMARY AND PROSPECTS has been changed to

We have demonstrated the control of the dissociation pathways of a H_2^+ molecule in a time scale of sub-10 fs using the vibrational dynamics in the $1s\sigma_g$ state of a H_2^+ molecular ion.

Finally, we are grateful to the reviewer for recognizing that manipulating vibrational and electronic wavepackets is sufficiently interesting to the specialist. Therefore, we would like the reviewer to notice the wording of the web page describing the content type of *Nature Communications*

(http://www.nature.com/ncomms/authors/content_types.html),

in which we can find the sentence, “*an Article is a novel and important research study of high quality and of interest to that specific research community*”.

This requirement for an Article coincides with the reviewer’s evaluation of our study, although we are confident that our study will attract the interest of a broad range of research communities. We hope that the reviewer will acknowledge our revised manuscript to be suitable for publication in *Nature Communications*.

Reviewer #2 (Remarks to the Author):

The authors have adequately addressed all my concerns. They have also satisfactorily addressed the comments of Referee 3, especially his/her point 8.

I recommend publication of this manuscript in its present form.

Reviewer #3 (Remarks to the Author):

Following modifications made in response to comments, I thank the authors for their efforts, which have improved the overall quality of the manuscript and its broader appeal. I now support the publication of the manuscript in Nature Communications and congratulate the authors on their achievements. The findings and presentation of results and calculations are in keeping with the requirements of the journal.